# Climatic modulation of surface acidification rates through summertime wind forcing in the Southern Ocean

Liang Xue [1,2], Wei-Jun Cai [3], Taro Takahashi[4], Libao Gao [1,2], Rik Wanninkhof[5], Meng Wei[1,2], Kuiping Li [1,2], Lin Feng[1,2] & Weidong Yu [1,2]

While the effects of the Southern Annular Mode (SAM), a dominant climate variability mode in the Southern Ocean, on ocean acidification have been examined using models, no consensus has been reached. Using observational data from south of Tasmania, we show that during a period with positive SAM trends, surface water pH and aragonite saturation state at 60°–55° S (Antarctic Zone) decrease in austral summer at rates faster than those predicted from atmospheric $CO_2$ increase alone, whereas an opposite pattern is observed at 50°–45° S (Subantarctic Zone). Together with other processes, the enhanced acidification at 60°–55° S may be attributed to increased westerly winds that bring in more "acidified" waters from the higher latitudes via enhanced meridional Ekman transport and from the subsurface via increased vertical mixing. Our observations support climatic modulation of ocean acidification superimposed on the effect of increasing atmospheric $CO_2$.

[1] First Institute of Oceanography, State Oceanic Administration, Qingdao 266061, China. [2] Laboratory for Regional Oceanography and Numerical Modeling, Qingdao National Laboratory for Marine Science and Technology, Qingdao 266237, China. [3] School of Marine Science and Policy, University of Delaware, Newark, DE 19716, USA. [4] Lamont–Doherty Earth Observatory of Columbia University, Palisades, NY 10964, USA. [5] NOAA Atlantic Oceanographic and Meteorological Laboratory, Miami, FL 33149, USA. Correspondence and requests for materials should be addressed to L.X. (email: xueliang@fio.org.cn) or to W.-J.C. (email: wcai@udel.edu)

The Southern Ocean has naturally low pH and saturation states of calcium carbonate ($CaCO_3$) due to cold temperatures and upwelling of $CO_2$-enriched deep waters, and it is vulnerable to ocean acidification (OA) caused by increasing atmospheric $CO_2$ levels[1–4]. Surface waters of the Southern Ocean are predicted to become undersaturated with respect to aragonite (a more soluble form of $CaCO_3$ relative to calcite) as early as year 2030 if sea surface $CO_2$ increases in concert with atmospheric $CO_2$ (ref. [3]). OA, defined as declining pH or $CaCO_3$ saturation states over decades or longer timescales[5], affects many marine organisms and especially fragile Southern Ocean ecosystems[6–8]. Although global OA is due primarily to increasing atmospheric $CO_2$ by fossil fuel combustion and land use changes since the Industrial Revolution[2,9], it may be enhanced by other processes such as upwelling, eutrophication, sea ice melt, and anomalous ocean circulation[10–17]. Such rapid acidification challenges the evolutionary adaptation capacity of organisms[18]. Therefore, understanding the processes or factors that modulate OA is important for projecting impacts on marine organisms and ecosystems.

Climatically, the Southern Ocean is sensitive, particularly during austral summer, to the Southern Annular Mode (SAM) that is the dominant mode of climate variability in the extratropical Southern Hemisphere[19,20]. This mode is quantified by the SAM index as the difference in normalized mean sea level pressure between 40° and 65° S (ref. [20]). In January, there was a positive SAM trend towards a high-index particularly since the 1980s, but this trend changed around 2000: the following decade exhibited decreased or no significant SAM trends (Fig. 1a). A positive SAM trend is associated with increasing westerly winds at high-latitudes (south of 55° S, Fig. 1b) resulting in increased equatorward Ekman transport and vertical mixing. Thus, "acidified" waters with lower pH and aragonite saturation state ($\Omega_{arag}$) from the south and from deeper depths are likely to be transported to the surface further north. Therefore, enhanced surface OA in excess of the effect of increasing atmospheric $CO_2$ may be expected at high-latitudes during a period with positive SAM trends. Here we define enhanced OA as evidenced by declining rates of pH or $\Omega_{arag}$ that are faster than rates predicted from increasing atmospheric $CO_2$ alone.

However, due partly to lack of observational data, previous studies on the effects of SAM on OA in the Southern Ocean use models which yielded different and even opposite conclusions[18,21–23]. Therefore, it is necessary and important to further investigate the mechanistic role of the SAM on sea surface carbonate chemistry and OA. Also, it is important to ascertain whether OA responds to the SAM differently for different latitudinal zones as was shown for circulation and biology[24], since the SAM measures a seesaw of atmospheric mass between the high-latitudes and mid-latitudes of the Southern Hemisphere[20].

Given that the region south of Tasmania is perhaps the only region where there is continuous observational $CO_2$ data since

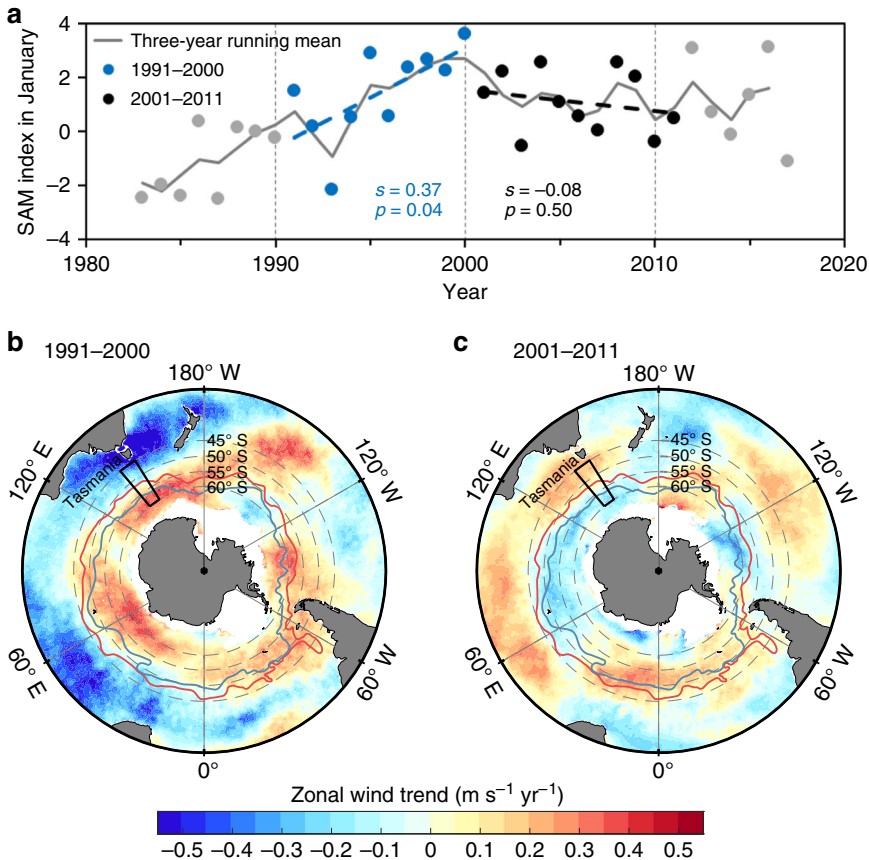

**Fig. 1** SAM index and change rates of zonal wind speed in the Southern Ocean. **a** SAM index in January 1983–2017 calculated by Marshall[20]. **b**, **c** Change rates of zonal wind speed in January 1991–2000 and January 2001–2011. In **a**, change rates of the SAM index (slope values) during the periods 1991–2000 (blue) and 2001–2011 (black) were determined using an ordinary least squares linear regression; slopes (s) and p-values of the regression analyses are also shown (differentiated with blue and black colors for the two periods). The gray line shows the weighted three-year running mean of the SAM index, which splits the data into two decades. In **b**, **c**, the red and blue lines show the mean positions of the subantarctic front (SAF) and the polar front (PF)[58], respectively; the black rectangle delineates the study area south of Tasmania. Change rates of zonal wind speeds, which are based on the CCMP wind product, were calculated using an ordinary least squares linear regression in each grid (0.25° × 0.25°)

1991 (Supplementary Fig. 1), we use observations from this area spanning two decades during 1991–2011, with contrasting SAM trends before and after 2000 (Fig. 1a and Supplementary Figs. 2–3) and show how changing wind patterns related to the SAM affect the rate of surface OA. We find that the SAM appears to have significant modulating effects on OA rates over different latitudinal zones. To account for the SAM modulation of OA rates, we examine mechanisms associated with wind-driven meridional Ekman transport and vertical mixing during austral summer when the upper ocean layers are stratified. Our work helps improve understanding of the mechanisms of OA in the Southern Ocean, thus providing observational constraints for the improvements of prediction models for ocean uptake of atmospheric $CO_2$ and impacts on the marine ecosystem.

## Results

**Changes of carbonate chemistry with time.** Using observed sea surface $CO_2$ fugacity ($f CO_2$), temperature (SST) and salinity (SSS) from the Surface Ocean $CO_2$ Atlas (SOCAT version 2)[25], and estimated total alkalinity (TA) from SSS, SST, and latitude (Fig. 2), we calculated dissolved inorganic carbon (DIC), pH, and $\Omega_{arag}$ over the two contrasting decades, 1991–2000 and 2001–2011 (see 'Methods'). The estimated values of TA and DIC agree well with measured data ('Methods' and Supplementary Fig. 4), giving high confidence in the calculated pH and $\Omega_{arag}$. To achieve a better spatial representation, prior to these calculations, the surface $f CO_2$, SST, and SSS data were binned and averaged within 0.02° latitudinal bands. Then averages were taken for the 5° latitudinal bands of 60°–55° S (high-latitudes or Antarctic Zone), 55°–50° S (transition zone or Polar Frontal Zone) and 50°–45° S (mid-latitudes or Subantarctic Zone)[26]. Finally, these data were adjusted to January values using the climatological seasonal variations described by Takahashi et al.[27] (see 'Methods'). While trends in SST, SSS, and TA were often not statistically significant, the relative rate of $f CO_2$ increase in surface water vs. that in the atmosphere was clear over the three regions and both time periods. A faster $f CO_2$ increase occurred during the pre-2000 positive SAM trend period in the high-latitude zone (60°–55° S), and a slower (or zero) increase in the mid-latitude zone (50°–45° S) compared to the atmospheric increase (Fig. 2).

Figure 3 shows that the rates of pH and $\Omega_{arag}$ change (i.e., rate of acidification) correlate with the SAM trends (Fig. 1a). At high-latitudes (60°–55° S), pH at in situ temperature (pH$_{@in~situ}$) decreased faster (0.0035 yr$^{-1}$) during the pre-2000 positive SAM trend than the pH decrease expected from atmospheric $CO_2$ increase alone (0.0020 yr$^{-1}$, gray dashed line, Fig. 3a). Correspondingly, $\Omega_{arag}$ at the in situ temperature ($\Omega_{arag@in~situ}$) decreased at a rate of 0.018 yr$^{-1}$, which is more than twice the rate of 0.007 yr$^{-1}$ due to atmospheric $CO_2$ alone (Fig. 3b). During the subsequent decade (2001–2011) when there was no significant SAM trend, pH$_{@in~situ}$ and $\Omega_{arag@in~situ}$ decreased at rates in accord with those predicted from atmospheric $CO_2$ (Fig. 3a, b).

In contrast, at mid-latitudes (50°–45° S), patterns opposite to those seen in the high-latitude band were observed (Fig. 3). During the decade of positive SAM trend (1991–2000), pH$_{@in~situ}$ decreased much slower than would be expected from atmospheric $CO_2$, and $\Omega_{arag@in~situ}$ even increased, although neither trend was statistically significant. During the subsequent decade (2001–2011) when there was no significant SAM trend, pH$_{@in~situ}$ and $\Omega_{arag@in~situ}$ both showed enhanced rates of decrease relative to the atmospheric $CO_2$ based prediction (Fig. 3i, j). For the transitional band (55°–50° S), the decrease in surface pH$_{@in~situ}$ during the two SAM periods was not statistically distinguishable from that predicted from atmospheric $CO_2$ and there were no significant changes in $\Omega_{arag@in~situ}$ (Fig. 3e, f). Overall,

acidification rates differ during different SAM-trend periods and within different latitudinal bands, similar to the responses of circulation and biology to SAM[24], suggesting that the influence of SAM on the acidification rates was likely associated with SAM-sensitive physical and/or biological factors.

**Correlation between wind trend and OA rates.** Our results display a consistently negative correlation between pH$_{@in~situ}$ (or $\Omega_{arag@in~situ}$) and wind speed, despite varying latitudinal responses of wind speed to the SAM trend (Fig. 3). In the high-latitude 60°–55° S band, wind speed increased significantly during the 1991–2000 positive SAM trend (Fig. 3c), when pH$_{@in~situ}$ and $\Omega_{arag@in~situ}$ decreased faster than expected from the atmospheric $CO_2$ increase (Fig. 3a, b). During a period with an insignificant change in SAM trends in 2001–2011 when wind speed decreased or did not change significantly, pH$_{@in~situ}$ and $\Omega_{arag@in~situ}$ declined at rates similar to those expected from the atmospheric $CO_2$ increase. In contrast, in the mid-latitude 50°–45° S band, during the period of positive SAM trends when winds decreased (Fig. 3k), pH$_{@in~situ}$ only decreased slightly and $\Omega_{arag@in~situ}$ increased somewhat (though not significantly, Fig. 3i, j), whereas during a period with an insignificant change in SAM trends when winds increased (Fig. 3k), pH$_{@in~situ}$ and $\Omega_{arag@in~situ}$ decreased evidently (Fig. 3i, j). For the transitional 55°–50° S band, there were no apparent changes in wind speed and, correspondingly, there was no enhanced acidification during the two periods of 1991–2000 and 2001–2011 (Fig. 3e–g).

We see more clearly the complex effects of wind on rates of pH and $\Omega_{arag}$ change by subtracting their rates of decrease due solely to atmospheric $CO_2$ increase from the observed rates of pH and $\Omega_{arag}$ change (see 'Methods'). It is clear, after removing the effects of atmospheric $CO_2$ increase, that the rates of pH and $\Omega_{arag}$ change are negatively correlated with change rates of zonal wind speed over the two periods and the three latitudinal bands (Fig. 4). That is, increasing winds enhance acidification.

**Modulations of Ekman transport and vertical mixing on OA.** Considering the correlation between enhanced pH and $\Omega_{arag}$ decreases and zonal wind speed changes (Fig. 4), and the lateral and vertical distributions of pH and $\Omega_{arag}$ in the Southern Ocean (Fig. 5), we explore the impacts on surface acidification from lateral transport and vertical mixing, both of which are influenced by wind speeds. Note in this section we used values at the regional mean temperature of 7.45 °C or pH$_{@7.45}$ and $\Omega_{arag@7.45}$ to examine the non-thermal influences of pH and $\Omega_{arag}$ although temperature influence (thermal influences) on these parameters was relatively minor (see 'Methods'). We considered that, among various drivers listed in Table 1, wind-driven lateral or Ekman transport was one of the important contributors to the trend in pH and $\Omega_{arag}$ changes relative to the atmospheric $CO_2$ increase. As shown in Fig. 5a, b, surface pH$_{@7.45}$ and $\Omega_{arag@7.45}$ in the Southern Ocean decreased poleward. At high-latitudes (60°–55° S) during the positive SAM trend of 1991–2000, the increase in westerly winds (westerly anomaly) enhances equatorward Ekman transport (Supplementary Fig. 5a), causing more waters with low pH$_{@7.45}$ and $\Omega_{arag@7.45}$ ("acidified" waters) from further south to be transported to this zone (Fig. 5a, b). This should result in further decreases in pH and $\Omega_{arag}$ (enhanced acidification, Fig. 3a, b; Supplementary Fig. 6a, b) in addition to those due to atmospheric $CO_2$ increase. In contrast, in the mid-latitude band (50°–45° S), westerly winds decreased (easterly anomaly) during a positive SAM trend, resulting in decrease in equatorward Ekman transport (i.e., anomalous poleward Ekman transport, Fig. 5; Supplementary Fig. 5a) and hence a slight

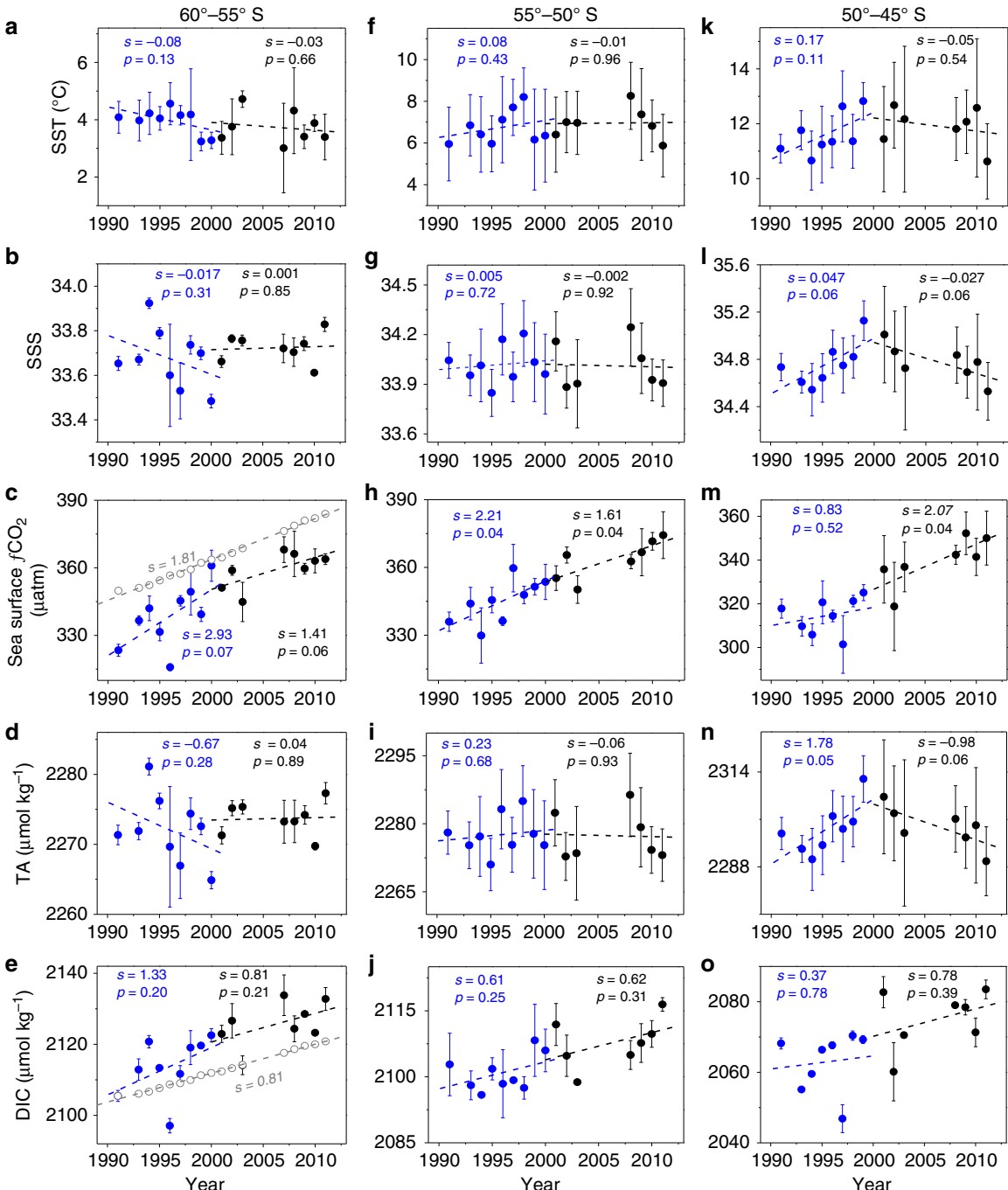

**Fig. 2** Temporal variability in sea surface temperature, salinity, and carbonate parameters in January in three latitudinal bands. **a–e** Sea surface temperature (**a**, SST), salinity (**b**, SSS), sea surface $CO_2$ fugacity (**c**, $fCO_2$), estimated total alkalinity (**d**, TA) and calculated dissolved inorganic carbon (**e**, DIC) at 60°–55° S. **f–j** Show the same parameters but at 55°–50° S; **k–o** also show the same parameters but at 50°–45° S (see 'Methods'). The vertical bars show one standard deviation, which reflects the spatial variability within each latitudinal band. Linear regression analyses were performed for the periods 1991–2000 (blue) and 2001–2011 (black). Slopes (s) and p-values of the regression analyses are also shown (differentiated with blue and black colors for the two periods). A trend of p-value < 0.1 is regarded as statistically significant (90% confidence interval) due to the small sample numbers (<10). Also, the atmospheric $CO_2$ data (shown as $fCO_2$) observed at the GCO (Cape Grim, Tasmania) atmospheric $CO_2$ measurement station (ftp://aftp.cmdl.noaa.gov/data/trace_gases/co2/flask/) and the DIC values computed due solely to the atmospheric $CO_2$ increase (see 'Methods') are indicated with open gray circles in Fig. 2c, e

increase in $pH_{@7.45}$ and $\Omega_{arag@7.45}$ (Supplementary Fig. 6k, l). This should counteract the acidification by increasing atmospheric $CO_2$, thus leading to no clear trends in $pH_{@in\ situ}$ and $\Omega_{arag@in\ situ}$ (suppressed acidification, Fig. 3i, j). Similarly, during the subsequent decade of 2001–2011 changes in $pH_{@7.45}$ and $\Omega_{arag@7.45}$ at high-latitudes and mid-latitudes (Fig. 3; Supplementary Fig. 6)

were also consistent with those expected by wind-driven Ekman transport (Supplementary Fig. 5b).

Since pH and $\Omega_{arag}$ decreased with depth (Fig. 5a, b), enhanced vertical mixing should also lead to an enhanced acidification. To determine whether changes in vertical mixing in the upper ocean can be a major contributor to a change of

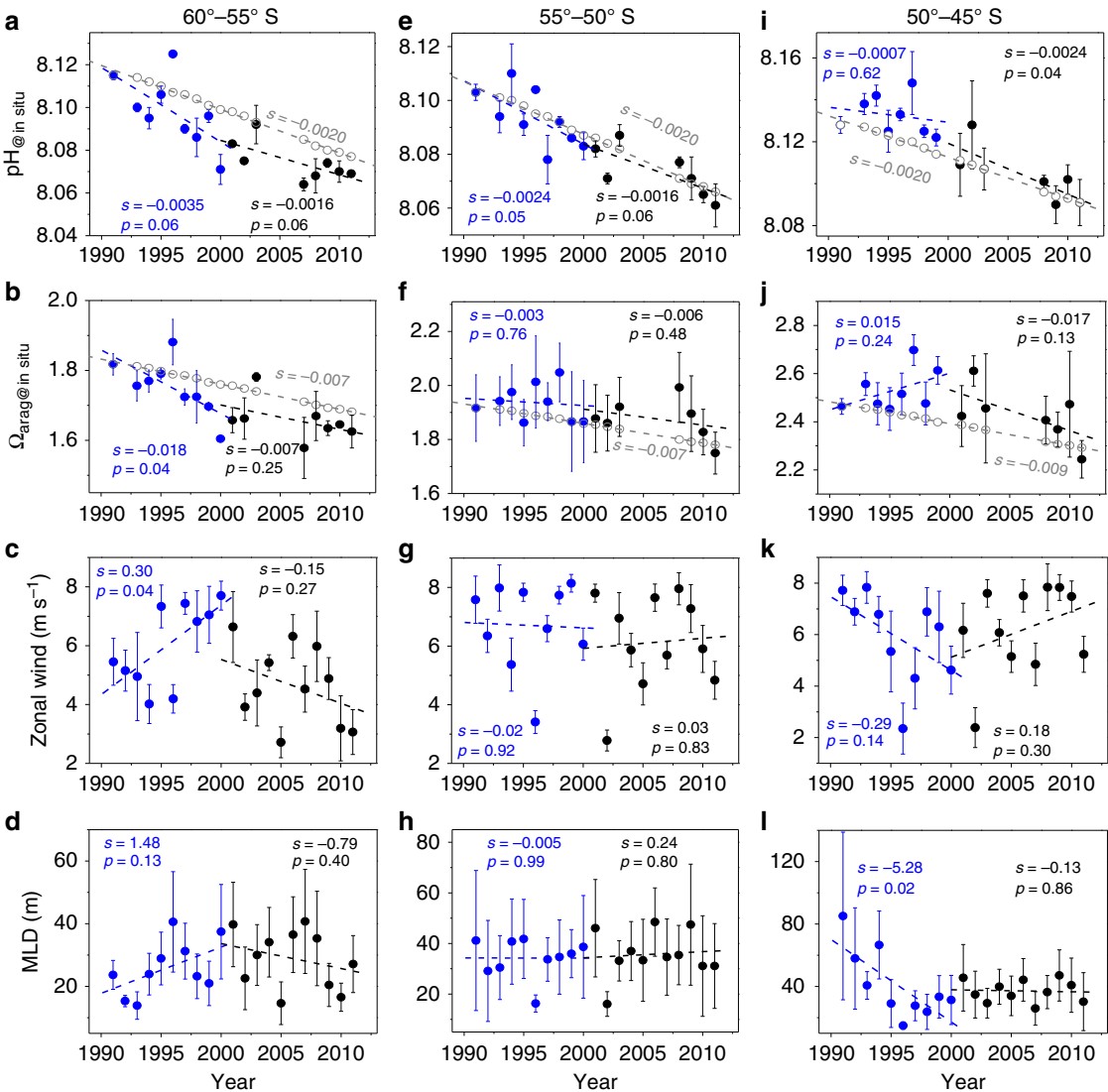

**Fig. 3** Temporal variability in sea surface pH, $\Omega_{arag}$, zonal wind speed and mixed layer depth (MLD) in January in three latitudinal bands. **a–d** Sea surface pH at in situ temperature (**a**, pH$_{@in\ situ}$), sea surface aragonite saturation state at in situ temperature (**b**, $\Omega_{arag@in\ situ}$), zonal wind speed (**c**) and mixed layer depth (**d**, MLD) at 60°–55° S. **e–h** show the same parameters but at 55°–50° S; **i–l** shows the same parameters but at 50°–45° S. The vertical bars show one standard deviation, which reflects the spatial variability within each latitudinal band. Linear regression analyses were performed for the periods 1991–2000 (blue) and 2001–2011 (black). Slopes ($s$), and $p$-values of the regression analyses are also shown (differentiated with blue and black colors for the two periods). Trends of $p$-value < 0.1 are regarded as significant statistically (90% confidence interval) due to the small sample numbers (<10). The open gray circles indicate the values computed due solely to the atmospheric $CO_2$ increase shown in Fig. 2c (see 'Methods'). Zonal wind speed and MLD are the mean values within 140°–148° E in the three latitudinal bands, respectively. Note by definition the trend of meridional Ekman transport is the same as that of zonal wind

acidification rates, we examined changes in mixed layer depth (MLD, Fig. 3d, h, l). During a period with positive SAM trends at high-latitudes (60°–55° S), MLD showed an increasing trend (Fig. 3d), suggesting an increase in vertical mixing that will entrain more subsurface waters with low pH and $\Omega_{arag}$ into the mixed layer (Fig. 5a, b), which enhances acidification rates. In contrast, in the mid-latitude band (50°–45° S), MLD showed a decreasing trend during a positive SAM trend (Fig. 3l), suggesting a decrease in vertical mixing that will entrain less subsurface waters into the mixed layer (Fig. 5a, b), which suppresses acidification rates. However, there were almost no changes in MLD during 2001–2011 at high-latitudes and mid-latitudes or in either period in the transition zone (Fig. 3), revealing that mixing in the upper ocean has no obvious changes during these periods.

It seems that Ekman transport brings more water from the higher latitudes than from the subsurface water to the surface Antarctic Zone, since observed changes of SST and SSS at the three latitudinal zones (Fig. 2) are consistent with changes expected due to Ekman transport (Supplementary Fig. 5a; Table 1). For example, at high-latitudes (60°–55° S) during a positive SAM trend, increased equatorward Ekman transport should induce a drop in SSS (Supplementary Fig. 7), whereas increased vertical mixing should cause a rise in SSS (Supplementary Fig. 7), but in fact we observed a decrease in SSS (Fig. 2b). However, vertical mixing may still play an important role in modulating OA due to the stronger gradients of pH and $\Omega_{arag}$ in the vertical direction than in the lateral direction (Fig. 5a, b; Supplementary Table 5). For instance, vertically from Point N to Point D, salinity increased by 0.08, and pH$_{@7.45}$ and $\Omega_{arag@7.45}$

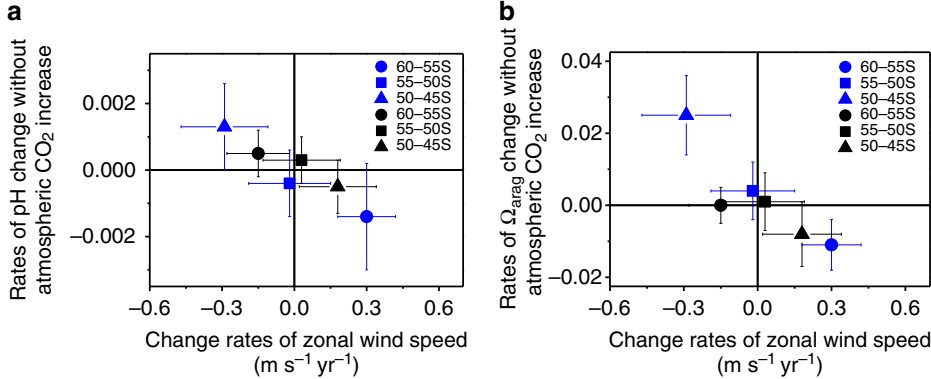

**Fig. 4** Impacts of SAM associated winds on acidification rates. **a**, **b** rates of surface pH (**a**) and $\Omega_{arag}$ change (**b**) without atmospheric $CO_2$ increase versus change rates of January zonal wind speed during two periods of 1991–2000 (blue) and 2001–2011 (black) in the three latitudinal bands of 60°–55° S, 55°–50° S and 50°–45° S. Rates of pH and $\Omega_{arag}$ change without atmospheric $CO_2$ increase are highly negatively correlated with change rates of zonal wind speed with a correlation coefficient of 0.92 and 0.89, respectively. Rates of pH and $\Omega_{arag}$ change without atmospheric $CO_2$ increase were the observed rates of $pH_{@in\ situ}$ and $\Omega_{arag@in\ situ}$ change, subtracting their rates predicted from atmospheric $CO_2$ increase alone (see 'Methods'). In this figure, negative change rates of pH or $\Omega_{arag}$ denote enhanced acidification compared to that predicted from atmospheric $CO_2$ increase alone. The bars show one standard deviation of change rates as shown in Fig. 3

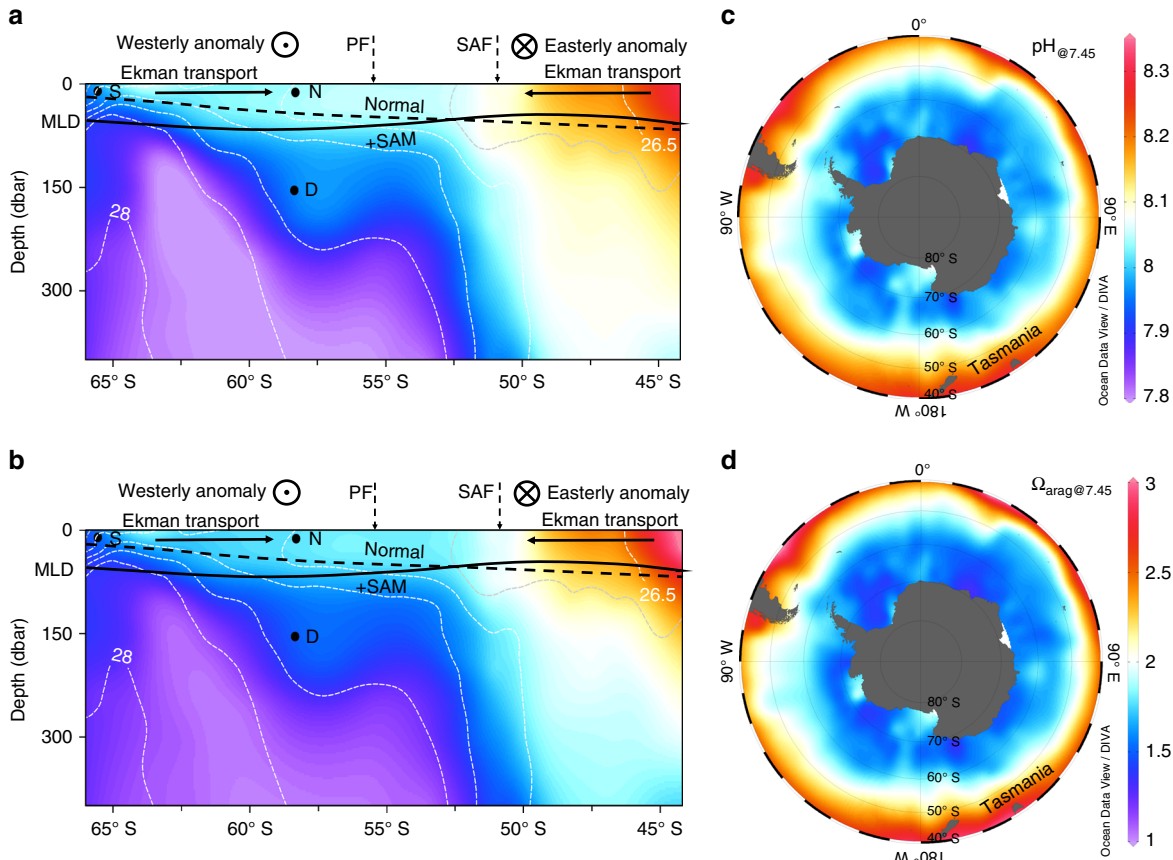

**Fig. 5** A schematic of Ekman transport and vertical mixing modulation of acidification rates as well as climatological distribution of surface $pH_{@7.45}$ and $\Omega_{arag@7.45}$. **a**, **b** Depict changes of wind-driven Ekman transport and mixed layer depth (MLD), and their influences on surface $pH_{@7.45}$ (**a**) and $\Omega_{arag@7.45}$ (**b**) in the region south of Tasmania during a positive SAM trend. There will be an anomalous equatorward Ekman transport when westerly winds increase (westerly anomaly), while there will be an anomalous poleward Ekman transport when westerly winds decrease (easterly anomaly). MLD is used for showing the changes in vertical mixing in the upper ocean. Black dashed (normal) and solid (+SAM) lines denote MLD before and during a positive SAM trend, respectively. In **a**, **b**, SAF and PF denote the mean positions of the subantarctic front (SAF) and the polar front (PF)[58]. In the water column, neutral density contours with an interval of 0.25 kg m$^{-3}$ ($\gamma^n$, white dashed line), and $pH_{@7.45}$ (**a**) and $\Omega_{arag@7.45}$ (**b**) distribution (shaded) observed along Transect SR03 during December 1994–January 1995 are shown. Also in **a**, **b** Points S, N, and D are shown, details about which can be found in Supplementary Table 5. In **c**, **d**, climatological distribution of surface $pH_{@7.45}$ (**c**) and $\Omega_{arag@7.45}$ (**d**) in January calculated from the TA and DIC data of Takahashi et al.[9] is shown. Note **a**, **c** use the same color bar, and **b**, **d** use the same color bar. Figure 5 is plotted using Ocean Data View (odv_4.7.10_w64 version)[59]

**Table 1 Observed trend change of OA, wind speed, Ekman transport, Ekman pumping velocity, mixed layer depth (MLD), sea surface temperature (SST), salinity (SSS), total alkalinity (TA), dissolved inorganic carbon gains due to air-sea $CO_2$ exchange ($\Delta DIC_{a-s}$) and chlorophyll _a_ in the three latitudinal bands of 60°–55° S, 55°–50° S, and 50°–45° S during the 1991–2000 positive SAM period**

| | 60°–55° S Antarctic Zone | 55°–50° S Polar Frontal Zone | 50°–45° S Subantarctic Zone | Note |
|---|---|---|---|---|
| Enhanced OA | + | nc | − | Fig. 3 |
| Wind speed | + | nc | − | Fig. 3 |
| Ekman Transport | + | nc | − | Suppl. Fig. 5 |
| Ekman Pumping | − | − | − | Suppl. Fig. 5 |
| MLD | + | nc | − | Fig. 3 |
| SST | − | nc | + | Fig. 2 |
| SSS | − | nc | + | Fig. 2 |
| TA | − | nc | + | Fig. 2 |
| $\Delta DIC_{a-s}$ | − | nc | + | Suppl. Fig. 6 |
| Chlorophyll _a_ | + | − | − | ref. [24] |

"+" denotes an increase with positive SAM trend, "−" denotes a decrease with positive SAM trend and "nc" denotes no change. Suppl. Fig. 5 and Suppl. Fig. 6 stand for Supplementary Figure 5 and Supplementary Figure 6, respectively. Note although SSS (or TA) change was not statistically significant (p-values of ~0.3) at high-latitudes (60°–55° S) probably due to small sample numbers of <10 (Fig. 2b, d), its trend of decrease was consistent with that observed in entire Antarctic Zone of the Southern Ocean[60]. Thus, in this table only trends with p-values > 0.3 are regarded as no change

decreased by 0.06 and 0.25 units, respectively, with $pH_{@7.45}$ change per unit salinity of −0.72 and $\Omega_{arag@7.45}$ change per unit salinity of −3.01. Laterally, from Point S to Point N, salinity, $pH_{@7.45}$ and $\Omega_{arag@7.45}$ increased by 0.42, 0.07 and 0.26 units, respectively, with $pH_{@7.45}$ change per unit salinity of 0.17 and $\Omega_{arag@7.45}$ change per unit salinity of 0.62. Therefore, vertical mixing could also play an important role in modulating OA. This is further supported by a mass balance model calculation (see 'Methods' and Supplementary Table 6). Overall, given the covariation of Ekman transport and vertical mixing with SAM associated winds (Table 1) and their consistent effects on pH and $\Omega_{arag}$, they both synergistically modulated the OA rates caused by increasing atmospheric $CO_2$.

Note in our paper we use vertical mixing rather broadly and mean to include convergence (i.e., downwelling) or divergence (i.e., upwelling), trend changes of which are quantified by Ekman pumping velocity (Supplementary Fig. 5c, d). We find there was a trend of decrease (increase) in Ekman pumping at the three latitudinal zones during 1991–2000 (2001–2011) (Supplementary Fig. 5c, d), but it appears that their influences on changes of pH and $\Omega_{arag}$ are minor or not observed. This can be seen, for example, from the transition zone (55°–50° S) where there were no apparent changes in wind speed, MLD or Ekman transport in either period (Fig. 3g, h; Supplementary Fig. 5a, b). During 1991–2000 when there was a tendency toward anomalous convergence (i.e., decreasing Ekman pumping, Supplementary Fig. 5c), waters with relatively low pH and $\Omega_{arag}$ from the high-latitude zone and waters with relatively high pH and $\Omega_{arag}$ from the mid-latitude zone may have been simultaneously transported to the transition zone (Fig. 5a, b), resulting in the cancellation of these effects and no net enhanced acidification during this period (Fig. 3e, f). While there was a tendency toward increasing upwelling during the period 2001–2011 (i.e., increasing Ekman pumping, Supplementary Fig. 5d), the influence of upwelling on SST, SSS, pH, and $\Omega_{arag}$ was not observed in the transition zone (Figs. 2 and 3). It indicates that the influence of upwelling on upper mixed layer was probably small and was susceptible to other processes, which needs further studies in this region.

Several processes could contribute to enhanced OA relative to atmospheric $CO_2$ increase. These could include an increase in equatorward transport of high-latitude water, an increase in vertical mixing/upwelling, a decrease in biological production, and an increased gas exchange rate due to increasing wind speed. While we cannot discount the importance of changes in local air-

sea $CO_2$ flux and biological activity, the mass balance analysis suggests that they both only play a relatively minor role in modulating acidification induced by increasing atmospheric $CO_2$ (see 'Methods'). Instead, the trend in TA (Fig. 2) and the mass balance analysis (see 'Methods') confirm that lateral and vertical transport are the dominant processes in modulating OA rates, since TA is immune to air-sea $CO_2$ fluxes and is just weakly influenced by biology[28]. By these analyses and by comparing observed changes of quantities with expected influences of SAM-associated winds (Table 1), we therefore conclude that the observed enhanced acidification in the high-latitudes and suppressed acidification in the mid-latitudes are primarily attributable to wind-driven Ekman transport and vertical mixing.

## Discussion

To put our findings in a broad context, we explore the possible influence of the SAM on pH and $\Omega_{arag}$ changes in other regions of the Southern Ocean by examining CCMP (Cross-Calibrated Multi-Platform) January zonal wind trends in the Southern Ocean basin (Fig. 1b, c). We find that during positive SAM trends (1991–2000), January wind speed increased at high-latitudes (poleward of the subantarctic front, SAF), and decreased at mid-latitudes (equatorward of SAF) except in parts of the Pacific sector, similar to the patterns shown by Lovenduski and Gruber[24]. In contrast, during a period with insignificant SAM trends (2001–2011), January zonal winds generally exhibited an opposite spatial pattern—decreasing at high-latitudes and increasing at most mid-latitudes. In January, across the entire Southern Ocean wind speed trends were largely consistent with those in our study region, though there were regional differences in the location of the transitional zone between high-latitude and mid-latitude patterns (Fig. 1b, c). This suggests that our study area can represent the meridional feature of the wind change in the Southern Ocean, and that the SAM modulation of OA rates south of Tasmania is part of a SAM-regulated change spanning the whole Southern Ocean. Given the spatial distribution of $pH_{@7.45}$ and $\Omega_{arag@7.45}$ in the entire Southern Ocean (Fig. 5), it may be inferred that during austral summer, the SAM would have substantial impacts on acidification rates in the whole Southern Ocean via wind forcing, although some heterogeneities do exist among the different Southern Ocean sectors. For instance, during 2001–2011, January zonal wind speed showed a decreasing trend in the 60°–55° S band south of Tasmania and in most other sectors in the Southern Ocean, but a weak increasing trend in the

Drake Passage (68°–56° W in Fig. 1c). Accordingly, the rate of $\Omega_{arag}$ decrease during this period appeared to be somewhat greater in the Drake Passage (0.013 ± 0.003 yr$^{-1}$ for austral summer during 2002–2015, see Munro et al.[29]) than in the high-latitude band in our study area (0.007 ± 0.005 yr$^{-1}$). This case further supports the mechanism of wind-driven modulation of OA, although it still needs to be verified in other sectors since the pattern of MLD change south of Tasmania is not fully consistent with that in other parts of the Southern Ocean (Supplementary Fig. 5e, f).

Our study based on observational data further supports the idea that the SAM has modulating effects on the Southern Ocean CO$_2$ system, mainly via wind-driven Ekman transport and vertical mixing during stratified austral summer (Supplementary Fig. 7). Studies based on ocean circulation models[30,31] and atmospheric CO$_2$ inverse methods[23] suggest modulation of the SAM on Southern Ocean carbon uptake. This conclusion is also supported by fCO$_2$ observations[32–34]. In contrast, using data derived by a neural network technique, Landschützer et al.[35] find that, on an annual basis, the reinvigoration of Southern Ocean CO$_2$ uptake after the early 2000s cannot be explained by the SAM-associated wind trends, because there are almost no changes in annual trends of wind between the 1990s and the 2000s (Supplementary Fig. 8). Instead, they propose a mechanism associated with a more zonally asymmetric atmospheric circulation[35]. More recently, however, using ocean circulation models, DeVries et al.[36] find that increased Southern Ocean CO$_2$ uptake in the 2000s compared to the 1990s was due to reduced upwelling (a weakened upper-ocean overturning circulation, similar to our mechanism).

There are at least two possible reasons that can explain why our results differ from those of Landschützer et al.[35]. One important reason is that there would be a large difference between seasonal trends (e.g., summer) and annual trends. To validate this, we chose the CCMP wind data[37,38] to examine the differences between January and annual trends. We find that there are substantial differences between January and annual trends (Fig. 1 and Supplementary Fig. 8). Another reason is that the SAM and its effects have a strong seasonality, with the most pronounced influence during austral summer[19,20]. Additionally, we recognize that changes in water properties of previous winter and spring seasons may also affect summer water properties, which is not discussed in our work due to data limitations and should be concerned in future observational and modeling efforts. Therefore, our study supports SAM modulation of acidification rates during austral summer (January), but sufficient observational data are not available[27] to elucidate the full annual influence of SAM on acidification. To resolve these issues, more observations are needed in this remote ocean for all seasons, especially during the poorly sampled austral winter.

Our work clarifies the discrepancy regarding the influence of the SAM on Southern Ocean acidification. Previous studies in this aspect are all based on models that yielded different understandings of SAM impacts. Some of these studies indicate that a positive SAM trend would not substantially affect OA or that the role of climate-driven physical changes would be minor[18,21]. While other studies argue that OA in the entire Southern Ocean would be enhanced during a positive SAM trend[22,23] in agreement with our viewpoint, they did not reveal latitudinal differences in their studies. Therefore, our work helps improve understanding of the mechanisms of OA in the Southern Ocean, which is important for modeling atmospheric CO$_2$ uptake and ecosystem responses. Overall, our work provides observational support for climatic modulation of OA, which should be taken into account in future predictions of acidification. It is most likely that climate change and variability have already been affecting the

advance of OA in the global ocean via wind forcing[14], which requires further observations.

## Methods

**Data collection and processing.** Sea surface fCO$_2$, SST, and SSS data from 17 cruises south of Tasmania (along or near Transect SR03, 60°–45° S) during 1991–2011 are used in this study (Supplementary Figs. 2–3). These data were extracted from the Surface Ocean CO$_2$ Atlas (SOCAT version 2)[25] (http://www.socat.info/) and were collected by the groups Bronte Tilbrook, Hisayuki Y. Inoue, Nicolas Metzl, Rik Wanninkhof, and Taro Takahashi[32,39–41]. TA and DIC data from discrete seawater samples collected along Transect SR03 during December 1994–January 1995, September 1996, March 1998, October–November 2001 and April 2008 in combination with salinity, temperature, and latitude data were used to derive the TA relationship (see 'TA estimation'). Also, TA and DIC data during December 1994–January 1995 were used to calculate pH$_{@7.45}$ and $\Omega_{arag@7.45}$ as shown in Fig. 5a, b. These data were obtained from the Global Ocean Data Analysis Project, Version 2 (GLODAPv2)[42].

SAM index used in this study is observation-based and developed by Marshall[20], and is available at http://www.antarctica.ac.uk/met/gjma/sam.html. CCMP wind data[37] were chosen for examining changes in zonal wind speed because of their good data quality[38]. This product has a resolution of 0.25° and is available at http://podaac.jpl.nasa.gov/datasetlist?search=ccmp. Monthly mean MLD data with a resolution of 0.5° determined by temperature criteria from SODA v3.3.1 (Simple Ocean Data Assimilation, available at http://apdrc.soest.hawaii.edu/dods/public_data/SODA/soda_3.3.1/) were used to examine the variability of MLD during January 1991–2011.

Before calculating carbonate system parameters, we binned, averaged, and deseasonalized the surface underway data, including sea surface fCO$_2$, SST, and SSS. For each parameter, we first binned all data points into 0.02° latitudinal bands to overcome different sampling frequencies among the cruises, calculated the average for each band, and finally took average values for the latitudinal bands of 60°–55° S, 55°–50° S and 50°–45° S, respectively as in Xue et al.[34]. For deseasonalization, we adjusted the averaged fCO$_2$, SST, and SSS data to January values using the long-term averaged seasonal cycle obtained by Takahashi et al.[27] (Supplementary Tables 1–3) as done by e.g., Lauvset and Gruber[43]. January was chosen because it is the month during which most data are collected (Supplementary Figs. 2–3) and also because the influence of the Antarctic ozone hole on surface climate is most pronounced during austral summer[19]. In addition, there is an evident SAM trend during this month (Fig. 1a). The averaged and deseasonalized fCO$_2$, SST, and SSS values at 60°–55° S, 55°–50° S, and 50°–45° S bands between 142.5° and 147.5° E are shown in Fig. 2.

**TA estimation.** Surface water TA within the study area was estimated using SSS (PSS), SST (°C) and latitude (Lat, in decimal degrees, negative for South latitudes) via Eq. (1):

$$TA\,(\mu mol\ kg^{-1}) = 35.94 \times SSS + 0.49 \times SST - 1.65 \times Lat + 964.97 \qquad (1)$$

($r^2 = 0.92$, $n = 346$)

Equation (1) was determined via multiple linear regression using the measured data in the upper 60 dbar along Transect SR03 collected during December 1994–January 1995, September 1996, March 1998, October–November 2001 and April 2008 (see 'Data collection and processing'). A comparison between estimated TA and the measured TA yielded a root mean square error (RMSE) of ±3.5 μmol kg$^{-1}$ (Supplementary Fig. 4a). This is better than what was initially derived from the global equation of Lee[44], which generates a RMSE of ±6.4 μmol kg$^{-1}$.

To examine uncertainty associated with the TA estimation, DIC, pH, and $\Omega_{arag}$ were calculated from fCO$_2$ derived from the observed TA and DIC and the estimated TA, using the CO2SYS program[45] and the apparent carbonic acid dissociation constants of Mehrbach et al.[46] as refit by Dickson and Millero[47]. The resulting RMSEs for DIC, pH, and $\Omega_{arag}$ were ±3.0 μmol kg$^{-1}$, ±0.0010 and ±0.005, respectively, when compared with measured DIC, and calculated pH and $\Omega_{arag}$ from a measured DIC and TA pair (Supplementary Fig. 4b–d). According to the error-calculation method of Lauvset and Gruber[43], the calculation errors of pH and $\Omega_{arag}$ are estimated to be 0.0022 and 0.010, respectively. Given the uncertainties of ±(0.005–0.01) for spectrophotometrically measured pH (refs. [48–50]) and ±0.18 for $\Omega_{arag}$ calculated from paired measurements of carbonate parameters[51,52], we conclude that error associated with the estimation of TA will not affect our results or conclusions.

**Calculation of pH and $\Omega_{arag}$.** Surface pH on the total H$^+$ concentration scale at in situ temperature and at the regional mean temperature of 7.45 °C was calculated using the CO2SYS program[45], with inputs of measured surface fCO$_2$ and estimated TA (Fig. 2) and climatological phosphate and silicate concentrations (Supplementary Table 4, though nutrient effects on pH and $\Omega_{arag}$ are small). The apparent carbonic acid dissociation constants of Mehrbach et al.[46] as refit by Dickson and Millero[47] were used, as recommended by Chen et al.[53] for polar ocean waters. For

calculating $\Omega_{arag}$ ($=[CO_3^{2-}] \times [Ca^{2+}]$ / $Ksp_{aragonite}$), carbonate ion concentration ($[CO_3^{2-}]$) was also calculated using the CO2SYS program[45]. The calcium ion concentration ($[Ca^{2+}]$) was calculated from salinity (0.01026 / 35 × salinity [mol kg$^{-1}$]) based on the conservative behavior of $[Ca^{2+}]$ to salinity[54], and the apparent solubility product of aragonite ($Ksp_{aragonite}$) was calculated after Mucci[52]. Also, $pH_{@7.45}$ and $\Omega_{arag@7.45}$ in the upper 400 dbar along Transect SR03 during December 1994–January 1995 (Fig. 5a, b) and climatological values of surface $pH_{@7.45}$ and $\Omega_{arag@7.45}$ in January (Fig. 5c, d)[9] were calculated from the TA and DIC data, respectively.

**Quantification of ocean acidification rates.** First, we calculated a weighted three-year running mean (1:2:1) for the SAM time series and found that the January SAM trend showed a clear shift in 2000 (gray line in Fig. 1a), splitting the data into two decades (i.e., 1991–2000 vs. 2001–2011). Then, following the definition of OA[5], rates of OA are characterized by rates of pH and $\Omega_{arag}$ change with time (i.e., slopes in Fig. 3), which were obtained using ordinary least squares linear regressions over each of the two decades of interest. To obtain the rates of decrease of pH and $\Omega_{arag}$ due solely to atmospheric $CO_2$ increase, we used the CO2SYS program[45] to calculate pH and $\Omega_{arag}$ from constant TA and increasing $fCO_2$, i.e., we held TA, SSS, and SST constant at their 1991 values while allowing surface water $fCO_2$ to increase at the same rate as atmospheric $CO_2$ observed at the GCO (Cape Grim, Tasmania) atmospheric $CO_2$ measurement station (ftp://aftp.cmdl.noaa.gov/data/trace_gases/co2/flask/; Fig. 2c). Similarly, changes in DIC and the difference between TA and DIC ([TA−DIC]) due solely to the increase in atmospheric $CO_2$ were calculated. Given that air-sea $CO_2$ exchange affects DIC but not TA, changes in [TA−DIC] that are due solely to changes in atmospheric $CO_2$ will have the same amplitude as the changes in DIC but will be of opposite sign (i.e., different direction of change). Rates of pH and $\Omega_{arag}$ change without atmospheric $CO_2$ increase (i.e., excluding the effects of increasing atmospheric $CO_2$, Fig. 4) were the observed rates of $pH_{@in\ situ}$ and $\Omega_{arag@in\ situ}$ change subtracting their rates predicted from atmospheric $CO_2$ increase alone (shown by gray dashed lines in Fig. 3).

**Thermal influences on OA rates.** The thermal influences on changes of pH and $\Omega_{arag}$ due to temperature changes are relatively minor. This is because there were no substantial changes in SST during the study period, although during a positive SAM trend (1991–2000) SST showed a decrease trend at 60°–55° S band, and an increase trend at 50°–45° S band (Fig. 2a, f, k). During 1991–2000 for the 60°–55° S band, although there was a decrease trend in SST of 0.08 °C yr$^{-1}$ (Fig. 2a), thus tending to increase pH (ref. [55]), the observed rate of $pH_{@in\ situ}$ decrease was still faster than the rate attributable solely to atmospheric $CO_2$ increase (gray dashed line, Fig. 3a). Comparing the rates of decrease for $pH_{@in\ situ}$ (Fig. 3a) and $pH_{@7.45}$ (Supplementary Fig. 6a) during this period shows that the effect of decreasing SST only partly counteracted the pH decreases. During 1991–2000 for the 50°–45° S band, the difference between the $pH_{@in\ situ}$ and $pH_{@7.45}$ trends is not statistically significant (Fig. 3i and Supplementary Fig. 6k). Compared to pH, $\Omega_{arag}$ is relatively insensitive to temperature changes, with surface $\Omega_{arag@in\ situ}$ and $\Omega_{arag@7.45}$ showing almost the same variability and rates of change throughout (Fig. 3 and Supplementary Fig. 6).

**Mass balance analysis of Ekman transport and vertical mixing.** For convenience and effectiveness of discussion, we introduce a combined property, the difference between the concentrations of TA and DIC, i.e., [TA−DIC]. Unlike pH and $\Omega_{arag}$, [TA−DIC] is a conservative quantity composed of two conservative parameters, and hence is suited for analysis of water mass mixing. The [TA−DIC] approximates closely the concentration of carbonate ions ($[CO_3^{2-}]$) by definition[56] and can be used as a proxy for pH and $\Omega_{arag@in\ situ}$[57]. In our dataset, [TA−DIC] correlates well with $pH_{@7.45}$ and with $\Omega_{arag@in\ situ}$, with a correlation coefficient of $r$ of nearly one (Supplementary Fig. 9). Thus, [TA−DIC] can be used to assess the changing effect of a process on OA rate.

To examine the relative contribution of Ekman transport vs. vertical mixing on OA rates, we use salinity and TA as conservative tracers to resolve changes in TA and DIC due to changes in these two processes. In the following, we take the case at high-latitudes during the 1991–2000 positive SAM trend as an example, and estimate the relative contribution of these two processes. We selected three points along transect SR03 to be compared including surface point S (South) and point N (North) and deep point D under N (Fig. 5 and Supplementary Table 5). To derive quantitatively the amounts of TA, DIC, and [TA−DIC] changes caused by changes in lateral transport and vertical mixing during a positive SAM period, we consider the changes of salinity (S) and TA in surface waters at point N:

$$\Delta S_E + \Delta S_V = \Delta S \quad (2)$$

$$\Delta TA_E + \Delta TA_V = \Delta TA \quad (3)$$

where the sign "Δ" denotes changes of a parameter; and subscripts "E" and "V" denote Ekman transport and vertical mixing, respectively. Note that the mass balances are built upon the changes of salinity and TA during the period but not on the absolute amount due to lateral and vertical transports. Also, we neglect the influence of change in precipitation–evaporation balance as its influence on TA

and DIC is small and similar, and thus its influence on [TA−DIC] is negligibly small.

Based on gradients per salinity change between Points S and N (lateral), and Points D and N (vertical) (Supplementary Table 5), we

$$\Delta TA_E = 50.95 \times \Delta S_E \quad (4)$$

$$\Delta TA_V = 84.34 \times \Delta S_V \quad (5)$$

Thus, Eq. (3) can be rewritten as

$$50.95 \times \Delta S_E + 84.34 \times \Delta S_V = \Delta TA \quad (6)$$

Since $\Delta S$ and $\Delta TA$ are known during the positive SAM period (slope values in Fig. 2b, d), through Eq. (2) and (6), we obtain $\Delta S_E = -0.023$ yr$^{-1}$ and $\Delta S_V = 0.006$ yr$^{-1}$ during the positive SAM period. Thus, based on the gradients of TA and DIC shown in Supplementary Table 5, the respective contribution of Ekman transport and vertical mixing on TA, DIC, and [TA−DIC] can be calculated (Supplementary Table 6

$$\Delta TA_E = -0.023 \times 50.95 = -1.17 \ \mu mol \ kg^{-1} \ yr^{-1},$$

$$\Delta DIC_E = -0.023 \times (-18.96) = 0.44 \ \mu mol \ kg^{-1} \ yr^{-1},$$

$$\Delta [TA-DIC]_E = -1.61 \ \mu mol \ kg^{-1} \ yr^{-1};$$

$$\Delta TA_V = 0.006 \times 84.34 = 0.51 \ \mu mol \ kg^{-1} \ yr^{-1},$$

$$\Delta DIC_V = 0.006 \times 366.27 = 2.20 \ \mu mol \ kg^{-1} \ yr^{-1},$$

$$\Delta [TA-DIC]_V = -1.69 \ \mu mol \ kg^{-1} \ yr^{-1}$$

Our calculations show that at high-latitudes during the 1991–2000 positive SAM trend the contribution of Ekman transport and vertical mixing on OA rates (as Δ[TA−DIC]) are likely on the same order of magnitudes (Supplementary Table 6).

**Impacts of air-sea $CO_2$ flux and biological activity on OA.** Similar to salinity and TA, we have the mass balance of DIC

$$\Delta DIC_E + \Delta DIC_V + \Delta DIC_A + \Delta DIC_B = \Delta DIC \quad (7)$$

Here "A" and "B" denote changes in air-sea exchange and biological activity, respectively.

Based on this mass balance and the observed $\Delta DIC$ and calculated $\Delta DIC_E$ and $\Delta DIC_V$, the total contribution to DIC change and thus [TA−DIC] change from air-sea gas exchange and biology is obtained (Δ[TA−DIC]$_A$ + Δ[TA−DIC]$_B$ = 1.30 μmol kg$^{-1}$ yr$^{-1}$), which is less than the effect by physical transports associated with changes in wind speed (Δ[TA−DIC]$_E$ + Δ[TA−DIC]$_V$ = −3.3 μmol kg$^{-1}$ yr$^{-1}$) (Supplementary Table 6). This result is consistent with that obtained by analyzing the trend changes of each process (Table 1). For example, during 1991–2000 at high-latitudes both decreasing air-sea $CO_2$ flux and increasing biological production[24] should result in a decrease in DIC (Supplementary Fig. 6; Table 1) and thus an increase in [TA−DIC], which can partly cancel out OA. In contrast, during this period increasing Ekman transport and vertical mixing (Table 1) should enhance OA. Comparing with observed enhancement of OA rates (Fig. 3a, b) indicates that Ekman transport and vertical mixing play a dominant role in modulating OA rates.

Further, Δ[TA−DIC]$_A$ and Δ[TA−DIC]$_B$ during 1991–2000 at high-latitudes can be calculated (Supplementary Table 6). When the time of 30 (or 100) days is considered for $CO_2$ uptake each summer, $\Delta DIC_A$ and Δ[TA−DIC]$_A$ would be −0.29 (or −0.97) μmol kg$^{-1}$ yr$^{-1}$ and 0.29 (or 0.97) μmol kg$^{-1}$ yr$^{-1}$, respectively (slope values in Supplementary Fig. 6e), and thus $\Delta DIC_B$ and Δ[TA−DIC]$_B$ would be −1.01 (or −0.33) μmol kg$^{-1}$ yr$^{-1}$ and 1.01 (or 0.33) μmol kg$^{-1}$ yr$^{-1}$, respectively (Supplementary Table 6). Despite the fact that we cannot fully constrain the relative contribution between air-sea $CO_2$ flux and biology, during the 1991–2000 positive SAM period at high-latitudes, biological carbon uptake induced an increase in [TA−DIC], reducing OA, which is consistent with the increasing biological production reported previously[24]. Note that in this paper we discuss the decadal changes of parameters or processes rather than seasonal changes of them. For example during 1991–2000 at high-latitudes the decrease in air-sea $CO_2$ flux (Supplementary Fig. 6d) and the increase in biological production[24] (characterized by chlorophyll) both should decrease DIC and thus increase [TA−DIC], reducing OA, although on seasonal timescale, for example,

during January air-sea $CO_2$ flux (absorbing $CO_2$) will increase DIC and reduce pH and $\Omega_{arag}$, and biological carbon uptake will decrease DIC and increase pH and $\Omega_{arag}$.

**Data availability**. Sea surface $fCO_2$ data can be obtained from the Surface Ocean $CO_2$ Atlas (SOCAT version 2) (http://www.socat.info/) and the data that support the findings of this study are available from the corresponding author upon reasonable request.

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

## Acknowledgements

This work was supported by the National Natural Science Foundation of China (41506099), the Basic Scientific Fund for National Public Research Institutes of China (2016Q01, 2018S02), the National Natural Science Foundation of China-Shandong Joint Fund (U1606405), the National Key R&D Program of China (2018YFA0605701), Chinese Projects for Investigations and Assessments of the Arctic and Antarctic (CHINARE2017–01–01, CHINARE2017–04–01, CHINARE2017–04–04) and Shandong Provincial Natural Science Foundation, China (No. ZR2016DQ02). We thank Drs. Gareth J. Marshall, Baochao Liu, Yongliang Duan, and Qi Shu for useful discussions. We acknowledge the efforts of the Surface Ocean $CO_2$ Atlas (SOCAT) in synthesis of the global surface ocean $CO_2$ data. SOCAT is an international effort, endorsed by the International Ocean Carbon Coordination Project (IOCCP), the Surface Ocean Lower Atmosphere Study (SOLAS) and the Integrated Marine Biogeochemistry and Ecosystem Research program (IMBER), to deliver a uniform quality-controlled surface ocean $CO_2$ database. The many researchers and funding agencies responsible for the collection of data and quality control are thanked for their contributions to SOCAT. Special thanks are given to Drs. Bronte Tilbrook, Claire Lo Monaco, Hisayuki Y Inoue, and Nicolas Metzl, who collected some of these data. W.-J.C. would like to thank the U.S. National Science Foundation (NSF), NASA, NOAA, and the University of Delaware (internal funds) for supporting his ocean carbon cycling research. T.T. was supported by a grant from the Climate Program Office of the National Oceanic and Atmospheric Administration (NOAA). R.W. was supported by NOAA's Office of Oceanic and Atmospheric Research (OAR) including the Ocean Observation and Monitoring Division of the Climate Program Office (fund reference100007298)

## Author contributions

L.X. and W.-J.C. designed this study and prepared the paper. T.T. and R.W. contributed parts of the data and writing. L.G., M.W., K.L., L.F., and W.Y. analyzed the data. All authors contributed to discussion and revision of the paper.

## Additional information

**Competing interests:** The authors declare no competing interests.

