## [Peer Review File · Nature Communications]

Reviewers' comments:

Reviewer #1 (Remarks to the Author):

Review of manuscript NCOMMS-17-11366-T EJones

Climate modulation of acidification rates in the Southern Ocean through wind forcing
Xue et al.

This study addresses the impact of the SAM on temporal changes in ocean acidification within different latitude bands of the Southern Ocean. Much remains to be resolved on the impacts of the SAM on CO₂ uptake and acidification, which is important for both observational and modelling studies in such a dynamic and climatically sensitive oceanic region. A suite of field observations from south of Tasmania is used to highlight the impact of a positive SAM period (1991-2000) on sea surface pH and aragonite saturation states in the Southern Ocean 55-60 °S. The positive SAM phase is associated with increasing westerly winds that drive northward Ekman transport and enhanced vertical mixing with low-pH deep waters in the south. This result was put into context by consideration of latitudinal heterogeneity and a following period of a contrasting (non-significantly positive or negative) SAM phase. The authors identified that a positive SAM trend was accompanied by decreases in surface water pH and aragonite desaturation with concomitant effect of enhanced rates of acidification of the Southern Ocean between 1991-2000.

The manuscript is well written with the range of oceanographic and atmospheric parameters presented and discussed in an interesting manner to explore the impacts of the SAM (with strong positive and insignificant phases identified) on acidification rates of the Southern Ocean. The data have been rigorously analysed, including internal consistency checks of the CO₂ system and having subsequent error estimates are applied, providing clear evidence for the interpretations and concluding statements. The discussions place the key conclusions within a context of latitudinal variations, temperature variations and relations to atmospheric CO₂ increases and consider the impacts of the results relative to those previously published. The text is supported by many elegant figures, both in the main manuscript and supporting material. This study highlights some new and significant results of wider appeal and adds valuable understanding to the climatic variability and controls on Southern Ocean CO₂ cycling and acidification and as such the results and concluding remarks should stimulate further discussion and sampling efforts and be considered in future studies in this field.

The choice of the journal fits very well and this article is recommended for publication. In the version reviewed here there are several minor ameliorations that should be made and the authors may find general and specific comments, questions and linguistic corrections listed below.

General:

1. Check the format of some literature references cited in the text, e.g., line 34, line 72, line 129.

Specific:

Line 32-37 long sentence, shorten for clarity

Line 35 "...bring low-pH waters..." to where? And add "...enhanced..." before "...wind-driven Ekman pumping..." in relation to influence of the SAM on acidification rates as Ekman pumping is a natural process that occurs irrespective of SAM phases

Line 35-36 re-phrase "...low-pH waters from higher latitudes and greater depths..." to "...low-pH deep waters at higher latitudes..."

Line 43 re-phrase to read "Surface waters of the Southern Ocean are..."

Line 44-45 re-phrase to read "...soluble form of CaCO₃ relative to calcite..."

Line 53 add a comma after "...Therefore..."

Line 54 replace "...important to our..." with "...important for our..."

Line 57 "...relatively abundant..." in both time and space? Clarify for what is relevant to the study

Line 58 "...rate of ocean acidification..." in what parts (depths) of the ocean? Specify surface water, upper ocean... for clarification

Line 61 replace "...gas..." with "...gases..."

Line 66 re-phrase to read "...enhanced upwelling of deep waters is expected to occur, supplying the surface with low-pH ..."

Line 68 replace "...(H..." with "...(h..."

Line 69 re-phrase to read "...evidenced by declining rates of pH and Ω_{arag} ..."

Line 70 rearrange to read "...alone)."

Line 77 replace "...confidence to..." with "...confidence in..."

Line 123-125 repetition in same sentence starting at "..., i.e., acidification was enhanced..." needs to be rephrased

Line 139-145 long sentence and poorly phrased at times, suggestion to split into 2-3 statements with a break at line 141 and line 143, for example

Line 142 rephrase to read "...waters with low (TA-DIC) from deeper depths and further south..."

Line 143 "...enhanced decrease..." can be a mis-leading coupling of positive-negative words, perhaps re-phrase to read "...result in further decreases in (TA-DIC)..."

Line 149 replace "...superimposition..." with "...superposition..."

Line 190 add a statement following "...reinvigoration of Southern Ocean CO₂ uptake..." that relates to the key results of this study, i.e., enhanced Ekman pumping and increased DIC in the surface ocean at southern latitudes, to better link it the impacts of SAM-driven winds and Southern Ocean CO₂ system

Line 192-194 reference to "... (supplementary Fig. 7)." Following "...SAM trend in January..." is inconsistent as supplementary Fig. 7 displays data from February and December, please check and amend accordingly and rephrase the sentence to be clear as to which month is being discussed for austral summer trends

Line 216 add "...ocean..." before "...acidification..." for more concise impact in the closing statements

Line 238 add "...the..." before "...World Ocean Atlas..."

Line 285 erroneous use of "...~...", please check

Line 311 rephrase to read "To obtain the rates of decrease of pH and..."

Line 342 what is the date of sampling for the surface water endmember values? This would have relevance for the high seasonality within the surface layer and include a statement for the justification of selecting a single value (consider a range, or mean and standard deviation)

Line 344 are the sea ice values means? perhaps with associated standard deviations? If this is available it would give an idea about variability of CO₂ system parameters within bulk sea ice

Line 348 replace "...observed..." with "...collected..."

Line 353 repetition of the word "...solely..."

Line 353-354 check the wording as currently this part of the sentence does not fully make sense

Line 361 add a few details to explain why "...TA decrease and DIC increase..." that would link into the detailed and informative paragraph above but help to make it more concise and bring together the concluding statements, e.g., "...TA decrease (decreases in salinity from northward Ekman drift) and DIC increase (increases in upwelling and subsequent mixing into CO₂-rich deep waters) ..."

Line 401 rephrase to read "...actual rate of decrease..."

Line 405 rephrase to read "Also, the effect of biological carbon uptake..." to make it more concise to which biological process is being discussed (not calcification) and directly relate to DIC concentrations and thus influence (TA-DIC)

Supplementary material

Figure 1 add units to colour bar for are fCO₂ values

Figure 3 (e) specify if this is measured or calculated DIC (as done for TA), for consistency and clarity

Figure 9 replace "...reginal..." with "...regional..."

Figure 10 replace "...reginal..." with "...regional..."

Table 2 add units to parameters in the caption

Reviewer #2 (Remarks to the Author):

This study aims to improve understanding of drivers of temporal variability in ocean acidification (OA) rate (and associated CO₂ chemistry changes) in the Southern Ocean, especially investigating the role of an atmospheric driver (i.e. Southern Annular Mode, SAM). This is an important topic with broad consequences that has been the focus of several recent studies so work like this could be influential. A plausible mechanism through which SAM could influence S. Ocean surface CO₂ chemistry is described (ie. +SAM => +winds => +vert. mixing => +OA at high latitudes; +SAM => -winds => -vert. mixing => depressed OA at low latitudes). Indeed there does appear to be an increasing trend in the January SAM index from 1990-2000 and then stable/slightly decreasing until 2010, and zonal wind fields are shown by the authors to be changing during those periods in the patterns that are described.

However, I have not been convinced that the CO₂ chemistry data support the second half of the mechanism anywhere near as strongly as claimed by the authors, for the following reasons. This is particularly important as this paper aims to change some previous thinking, for example disagreeing with Landschuetzer et al. (2015) on the influence of SAM-driven wind trends on CO₂ chemistry.

The first thing to keep in mind is that pH, omega and (TA-DIC) are closely related to each other through the CO₂ chemistry reactions on a fundamental level. They are therefore absolutely not independent from each other as lines of evidence, in terms of comparing their trends with the SAM index. Given this, and the way that the analysis has been performed – by splitting the data into two decades – we are left with only 2 data points (ie. rate of change of the CO₂ chemistry for 1990-2000 vs that for 2000-2010) with which the hypothesis can be tested. [As a general point I do not see much value in calculating and discussing (TA-DIC). The authors say it is correlated/roughly equal to other CO₂ chemistry variables of interest, but why not simply calculate those directly?]

Following from this, the authors seem to have implicitly assumed that the CO₂ chemistry response to SAM occurs on decadal timescales, but they haven't explained why. I see no reason why the mechanism that the authors describe should not also have an effect on the CO₂ chemistry on shorter, interannual timescales. Particularly given that the SAM index varies from year to year sometimes by amounts greater than the 10-year trend – so why don't we see a CO₂ chemistry response to these big changes on a year-by-year basis? Why is the relatively small long-term trend more important than the greater short-term variability for this apparently fast-operating process? Considering shorter term variability would be an easy way to add extra data points to the analysis, as no reason is given why that should be expected to drive the CO₂ chemistry any differently than the decadal SAM trends.

The choice of 2000 to split the analysis between increasing/decreasing SAM seems a little arbitrary, and I am not convinced that this 'break point' happens at the same time in the CO₂ chemistry data – does it in fact precede SAM? I would prefer to see an objective statistical break-point analysis, applied to both the SAM time series and the CO₂ chemistry data, in order to see if a genuine, robust change in the trend happens in both datasets at the same time. Alternatively/as well, a lot of work has been done recently on time of emergence in marine biogeochemical datasets, looking at how long a time-series is required to determine a reliable longterm trend given the variance of short-term variability. It would be good to see this methodology applied to see whether the 'trends' actually are taken over a period sufficient to reliably emerge from the background variability.

Why is the Jan SAM index value so much more important than Dec or Feb? There is no increasing/decreasing pattern at all for Dec and Feb as shown in the supplementary figs. Unless

there is a mechanistic explanation, this could be interpreted as evidence that any apparent correlation between SAM index trends and these CO₂ chemistry trends could be a fluke.

Reviewer #3 (Remarks to the Author):

Review of "Climatic modulation of acidification rates in the Southern Ocean through wind forcing"

Summary: The authors calculate surface pH and aragonite saturation state (basically TA-DIC) from measurements of sea-surface temperature, sea-surface salinity, and fCO₂ along a meridional transect in the Southern Ocean south of Tasmania, over the period 1990-2011. They compared trends in January (mid-summer) surface pH and aragonite saturation state over the period 1990-2001, during a positive phase of the Southern Annular Mode (SAM), and over the period 2001-2011 during a more neutral phase of the SAM. They found a negative correlation between the acceleration of zonal winds and the anomalous (atmospheric CO₂-detrended) surface pH and aragonite saturation state both south of the Antarctic Polar Front (APF) and north of the Sub-Antarctic Front (SAF), during both phases of the SAM. However, there was no apparent impact of wind-driven changes on pH or acidification state in the zone between the APF and the SAF. The authors conclude that ocean circulation changes, driven by changes in the strength of the westerly winds, has driven the observed anomalous trends in surface pH and aragonite saturation state.

The authors' analysis was generally okay, although they should propagate errors in the TA and DIC calculations into their regression analysis. The authors should also investigate the sensitivity of their trends to the chosen end points of their regression analysis. Since the reviewer instruction asked for it, I will also comment that the use of p-values for regressions is, in my opinion, only useful for controlled experiments and not environmental data where nothing is controlled. Nonetheless p-values are widely used and I do not necessarily object to their use here.

The major issue I have with the study is that I cannot entirely agree with its conclusions (although I am sympathetic to the general idea of circulation changes driving variability in Southern Ocean acidification rates). This is because if circulation variability drives changes in the surface pH and saturation state (i.e. TA-DIC) along the transect, then the region between the PF and the SAF should be sensitive to these changes. Looking at figure 1, it is clear that there is anomalous convergence (=downwelling) between 50-55 S in the study region during 1991-2000, and anomalous divergence (=upwelling) between 50-55 S in the study region during 2001-2011. Given the strong gradients in (TA-DIC) (Figure 4) one would expect these circulation-driven changes to have an impact in that region. But they do not, which is a puzzle.

Perhaps the more parsimonious explanation is that the changes in wind speed drive changes in air-sea CO₂ fluxes south of the PF, and north of the SAF, but not between the two where the wind speed is relatively constant (Figure 2). But the lack of expected dynamical response between the PF and the SAF is still puzzling.

Basically the authors need to provide a model (more sophisticated than their current conceptual mixing model) that explains why the region between the SAF and the PF is not responsive to the wind stress changes and the resulting changes in wind stress curl and Ekman suction in that region, whereas the other regions are.

Minor points:

The authors often use "mixing" when they appear to mean "upwelling". Or perhaps in some cases they really do mean mixing, in which case they should clarify.

The authors appear to discount changes in air-sea CO₂ fluxes as biology as important without any

particular explanation other than that another study found this to be so (on different time/space scales). Again, a more rigorous model would help with the attribution.